# Insights of Fe_2_O_3_ and MoO_3_ Electrodes for Electrocatalytic CO_2_ Reduction in Aprotic Media

**DOI:** 10.3390/ijms232113367

**Published:** 2022-11-02

**Authors:** Néstor E. Mendieta-Reyes, Alejandra S. Lozano-Pérez, Carlos A. Guerrero-Fajardo

**Affiliations:** Departamento de Química, Facultad de Ciencias, Universidad Nacional de Colombia, Bogotá 11321, Colombia

**Keywords:** CO_2_ reduction, TMO electrodes, electrocatalysis, dissociative electroreduction model

## Abstract

Transition metal oxides (TMO) have been successfully used as electrocatalytically active materials for CO_2_ reduction in some studies. Because of the lack of understanding of the catalytic behavior of TMOs, electrochemical methods are used to investigate the CO_2_ reduction in thin-film nanostructured electrodes. In this context, nanostructured thin films of Fe_2_O_3_ and MoO_3_ in an aprotic medium of acetonitrile have been used to study the CO_2_ reduction reaction. In addition, a synergistic effect between CO_2_ and the TMO surface is observed. Faradic cathodic processes not only start at lower potentials than those reported with metal electrodes, but also an increase in capacitive currents is observed, which is directly related to an increase in oxygen vacancies. Finally, the results obtained show CO as a product of the reduction.

## 1. Introduction

The current atmospheric levels of carbon dioxide and the depletion of fossil fuel reserves raise serious concerns about the resulting effects on the global climate change and on the future of some energy sources [1]. The electrochemical conversion of carbon dioxide is a promising way to address both problems simultaneously; this strategy provides an opportunity for the supply of platform chemicals, which can be used as supplementary raw materials in several industrial processes, or energy sources [2]. A wide variety of methods, such as photocatalysis, electrocatalysis, and heterogeneous catalysis, have all been successful in CO_2_ reduction [3]. Factors such as solvent, electrolyte, pH, temperature, CO_2_ concentration, electrode material, surface structure, and electrode potential influence the variety of products obtained [4,5]. Reduction products, including CO, CH_4_, and CH_3_OH, among others, have been reported [6,7,8]. This leads to the development of several reaction mechanisms with different numbers of electrons transferred, which determines the final oxidation state of the carbon atom [9]. However, due to the linear geometry with a closed-shell electronic configuration, the high energy level of its LUMO, and the low electron affinity of the molecule (0.6 eV), the reduction of CO_2_ is a non-spontaneous reaction requiring an energy input. Due to the repulsion between the newly acquired electron and the free electron pairs on the oxygen atoms, the transfer of an electron results in the formation of the anion radical CO_2_^(.−)^, which causes the molecular structure to bend [10,11,12,13].

In this sense, several reports have shown that the stabilization of the CO_2_^(.−)^ anion radical is an important step for achieving an efficient reduction of CO_2_. The CO_2_ molecule can then be activated on a solid-state catalyst, whose function is to adsorb the CO_2_ molecules by facilitating electron transference from the catalyst, providing favorable conditions for the bending of the molecule and the desorption of the products [10,11,14,15]. In this regard, the adduct formation of CO_2_^(δ.−)^ on different metal oxides has been widely reported [16,17].

On the other hand, in an aqueous solution, different factors, such as acid-base equilibria and hydrogen synthesis occurring as a competitive process, act against obtaining high efficiency and selectivity. In this sense, acetonitrile (ACN) is one of the most widely used non-aqueous aprotic solvents in catalysis and electrochemistry, and has been used to study the electrocatalytic CO_2_ reduction [18,19,20]. Pons et al. studied solutions of tetra-n-butylammonium tetrafluoroborate (TBAF) in dry ACN, finding ACN decomposition at potentials more negative than –3.0 V vs. Ag/Ag+ (0.01 M) [5]. The ACN solution containing tetrabutylammonium salts, such as the electrolyte system, TBAF (0.1 M), has been used for the study of the electrocatalytic reduction of CO_2_. This non-aqueous system not only offers stability at a potential window more negative than −1.97 V vs. SHE, which is needed for the first CO_2_ electron transfer, but also that avoids the hydrogen synthesis as a competitive reaction.

Metal oxides are used as electrocatalysts for CO_2_ reduction [21], and transition metal oxides (TMO) have been theoretically predicted to have efficient active sites for said reduction reaction [22]. In fact, these materials have reactive electrons, empty d orbitals, and crystalline defects, such as oxygen vacancies (OV), grain boundaries, and non-stoichiometry, that are capable of triggering the CO_2_ molecule activation, inducing the formation of the CO_2_^(δ·−)^ adduct [11]. Compared with metal catalysts, TMO can further improve the performance of CO_2_ reduction, which has drawn attention in recent studies. Some researchers have proposed that the active sites for CO_2_ in metal oxides are due to the reconstruction of their surface, formed by the reduced surface of the metal oxide at highly negative operating potentials [23], whereas others have highlighted metal-oxygen sites as active, leaving an open debate on this topic [24]. In this context and based on the dissociative reduction model previously reported for CO_2_ conversion at WO_3_ and TiO_2_ electrodes, results on Fe_2_O_3_ and MoO_3_ electrodes are shown for the CO_2_ electrochemical reduction. These findings provide an explanation for the electrocatalytic activity seen in other TMOs that have been previously studied, as well as their relationship with surface processes and the formation of OV, in terms of cathodic current and the initial potential that is related to CO_2_ reduction [25,26,27].

## 2. Results and Discussion

Fe_2_O_3_ and MoO_3_ thin-film electrodes were coated on Fluorine-doped Tin Oxide (FTO) substrates following two different synthesis routes, as explained in the Appendix A, leading to different morphologies. Synthesized Fe_2_O_3_ thin-film surface characterization is shown as to hematite, and a highly coated nanostructured electrode surface displaying 50 nm-thick nanoparticles is observed (Appendix A). An oxide film cross-section image, with approximately 500 nm-thickness, shows partially oriented nanobars. FE-SEM images for MoO_3_ electrodes, at two different magnification levels of 5000× and 25,000×, are shown in Appendix A. Electrodes prepared by sol-gel process are thoroughly coated, with no uncoated substrate areas (Appendix A). The morphology of the nanoparticles (Appendix A) is in accordance with electrodes synthesized by the sol-gel technique. The observed nanoparticles’ diameter is between 20–40 nm, with crystal aggregates well interconnected with each other. The X-ray diffraction pattern of the FTO-coated thin-film electrodes presented in Appendix A indicates that the as-prepared product presents high-crystallinity.

For electrochemical studies, the prepared electrodes were evaluated in a 0.1 M tetrabutylammonium perchlorate (TBAP)/ACN solution as the supporting electrolyte. The electrolyte solution was purged with CO_2_ and N_2_ for 30 min before the electrochemical procedure. Figure 1 shows the voltametric response for both the Fe_2_O_3_ and MoO_3_ working electrodes. The potential window for CO_2_ reduction was determined by purging with N_2_. Both Fe_2_O_3_ and MO_3_ electrodes show an increase in the cathodic current, with an onset at around −0.4 V and −0.6 V, respectively, which coincides with the beginning of the accumulation region. From a chemical point of view, such an increase can be seen as corresponding to Fe_(ss)_(III)/(II) and Mo_(ss)_(VI)/(V) redox processes in surface sites for Fe_2_O_3_ and MoO_3_ (indicated on the figure). Similar behavior has been observed in semiconductor materials such as TiO_2_ [28].

It has been reported that the metal reduced species in TMOs could be stabilized by small cations, such as H^+^ or Li^+^, through intercalation processes by local bronze formation [29]. However, in our case, one should bear in mind that the tetrabutylammonium cation (TBA^+^) is bulkier than the cavity available in the crystal lattice of metal oxide. Thus, TBA^+^ is sterically obstructed for intercalation. This fact not only acts in favor of OV formation, but also, the stabilization of the Fe_(ss)_(II) and Mo_(ss)_(V) metal surface centers with high electron density is not efficient. This is according with the previously reported dissociative electroreduction model [30]. Under these conditions, there is an increase in surface states energetically located close to the conduction band, and the OV formation is represented by the following equations:(1)$$\left(1+x\right){\mathrm{MoO}}_{3}+2x\;e\;\to {\;\mathrm{MoO}}_{3-x}+{\mathrm{xMoO}}_{4}^{2-}$$
(2)$$\left(1+x\right){\mathrm{Fe}}_{2}O_{3}+2x\;e\;\to {\;\mathrm{Fe}}_{2}O_{3-x}+{\mathrm{xFe}}_{2}O_{4}^{2-}$$
(3)$$\left(1+x\right){\mathrm{Fe}}_{2}O_{3}+2x\;e\;\to {\;\mathrm{Fe}}_{2}O_{3-x}+2{\mathrm{xFeO}}_{2}^{-}$$

In this dissociative model, the partial electroreduction of MO_3_ and Fe_2_O_3_ is limited to the interfacial region, whereas no significant structural damage occurs in the underlying region of the metal oxide film. According to Equations (1) and (3), the partial electroreduction of Mo and Fe lattice atoms from Mo^6+^ to Mo^5+^ and Fe^3+^ to Fe^2+^ induces the formation of oxygen-deficient metal oxide, ${\mathrm{MoO}}_{3-x}$ and ${\mathrm{Fe}}_{2}O_{3-x}$. Additionally, this mechanism entails a loss of mass due to the electrodissolution of MoO_3_ and Fe_2_O_3_ by the formation of ${\mathrm{MoO}}_{4}^{2-}$ and ${\mathrm{FeO}}_{2}^{-}$ [31]. 

Under the CO_2_ purged condition, faradaic currents at −0.6 V and −0.8 V are observed for Fe_2_O_3_ and MoO_3_, respectively. In both cases, the presence of CO_2_ increased the electrode capacitance, even at potentials where CO_2_ reduction does not occur. This fact suggests important changes in the electrode state that could be associated with an increase in the surface density of Mo^+5^- and Fe^+2^-reduced species as a result of the enhancement of the OV generation. This makes the potential at the onset of the reduction of metal atoms at TMO and the cathodic current associated with CO_2_ reduction indistinguishable. The increase in the capacitive current has been explained in terms of the increment of OV promoted by increasingly negative applied potential, which favors CO_2_ electro adsorption according to the equations:(4)$$\left(1+y\right){\mathrm{MoO}}_{3}+{\mathrm{yCO}}_{2}+2y\;e\;⇌{\;\mathrm{MoO}}_{3-Y}{\left({\mathrm{CO}}_{2}\right)}_{y}+{\mathrm{yMoO}}_{4}^{2-}$$
(5)$$\left(1+y\right){\mathrm{Fe}}_{2}O_{3}+{\mathrm{yCO}}_{2}+2y\;e\;⇋{\;\mathrm{Fe}}_{2}O_{3-Y}{\left({\mathrm{CO}}_{2}\right)}_{y}+{\mathrm{yFe}}_{2}O_{4}^{2-}$$
(6)$$\left(1+y\right){\mathrm{Fe}}_{2}O_{3}+{\mathrm{yCO}}_{2}+2y\;e\;⇋{\;\mathrm{Fe}}_{2}O_{3-Y}{\left({\mathrm{CO}}_{2}\right)}_{y}+2{\mathrm{yFeO}}_{2}^{-}$$

The adsorption of CO_2_ would, thus, promote the generation of non-stoichiometric Fe_2_O_(3−y)_(CO_2_)_y_ and MoO_(3−y)_(CO_2_)_y_. In this sense, the adsorption of CO_2_ induces the stabilization of reduced metal sites. The results show that CO_2_ binds slightly more strongly to stoichiometrically Ti(III)-deficient sites than to Ti(IV) sites, according to CO_2_ adsorption studies on TiO_2._ Therefore, it is reasonable to assume that CO_2_ adsorption is favored at reduced metal sites (Mo(V) and Fe(II)). Thus, a synergistic effect between CO_2_ and surface sites is verified. CO_2_ adsorption not only activates the molecule by the adduct formation, but also induces a promotion of OV and the electron-rich center formation in reduced metal species. By further reducing the applied potential, the resulting high concentration of electrons located in the near-surface region would facilitate charge transfer from the electrode to the adsorbed CO_2_ molecules, leading to the appearance of faradaic currents. It is worth noting that CO has been detected by gas chromatography analysis (presented in Appendix A), and the formation proceeds with two electrons transferred, according to Equations (7) and (8):(7)$$\left(1+y\right){\mathrm{MoO}}_{3-y}{\left({\mathrm{CO}}_{2}\right)}_{y}+2ye\;\to {\mathrm{MoO}}_{3-y}+\mathrm{yCO}+{\mathrm{yMoO}}_{4\;}^{2-}$$
(8)$$\left(1+y\right){\mathrm{Fe}}_{2}O_{3}+{\mathrm{yCO}}_{2}+2y\;e\;⇋\;Fe_{2}O_{3-y}+\mathrm{yCO}+{\mathrm{yFe}}_{2}O_{4}^{2-}$$

In the positive-going scan, a pseudocapacitive peak related to the reoxidation of the reduced metal is observed, and an enhancement in the electrode capacitance related to the increase in reduced metal sites Fe_(ss)_(II)/(III) and Mo_(ss)_(V)/(VI) is evident. Moreover, in the case of Fe_2_O_3_, an additional peak, probably related to the reoxidation of some CO_2_ reduction product, appears at 0.3 V. This fact is intriguing because it demonstrates that each metal oxide has unique electrocatalytic behavior for CO_2_ reduction and the formation of reduction products. It is important to highlight that carbon dioxide reduction does not proceed when a bare FTO conducting glass electrode is used.

To determinate the nature of the currents associated with CO_2_ electrocatalytic reduction, a linear sweep voltammetry (LSV) was performed. At a relatively low scan rate (5 mV·s^−1^), the capacitive current decreases, and the faradic currents are mainly observed. Figure 2 shows the initial potential at which the cathodic current for CO_2_ reduction appears, providing further evidence on the faradaic nature of the cathodic currents linked to CO_2_ reduction. However, the presence of CO_2_ generates changes in current profiles, and not only produces faradic currents due to CO_2_ reduction, but also increases capacitive currents due to the formation of electron-rich centers in reduced metal species. Therefore, to determinate the potential at which the faradaic process starts for the metal oxide materials studied, a double potential step chronoamperometry experiment was performed. A potential step was performed from the 1.0 V as initial potential (Pi), at which no faradaic process occurs, to several negative final potentials (Pf). Each step was applied for 59 s.

The chronoamperometric profiles shown in Figure 3 indicate the potential steps associated with each thin-film electrode. For MoO_3_, when the applied potential returns to Pi, the current in both CO_2_ and N_2_ purged is the same. In contrast, after electrode polarization to the initial potential from each final potential with CO_2_ electrode saturation, Fe_2_O_3_ shows an increase in anodic current. This could possibly be related to a reduction product reoxidation. In this sense, it is worth nothing that a wide potential peak at 0.2 V is observed, as can be seen from the cyclic voltammetry results. When a negative potential was applied under N_2_ and CO_2_ purging conditions, the currents were not always the same at the end of each step potential. For Fe_2_O_3_, an increase in current is observed from −0.3 V after 59 s under CO_2_ purge conditions, the increase being higher at −0.5 V. For MoO_3_, said increase in current becomes evident at −0.7 V.

Under CO_2_ and N_2_ bubbling conditions, the absolute value of the difference between observed currents as a function of the applied potential step is defined by Equation (9):(9)$$\left|j_{{\mathrm{CO}}_{2}\max-}j_{N_{2}\max\;}\;\right|\left(V\right)=\left|j_{{\mathrm{CO}}_{2}\max\left(v\right)}\right|-\left|j_{N_{2}\max\left(v\right)}\right|$$
where |j_(CO2)max(V)_| and |j_(N2)max(V)_| are the cathodic currents obtained 59 s after the electrode polarization occurs at the potential Pf.

A current profile calculated using Equation (9) has an increase in the relationship between faradaic and non-faradaic currents, as shown in Figure 4. In each case, the intercept with the potential axis corresponds to the onset potential for faradaic current related with the CO_2_ reduction. The onset for electrodes used in this study was −0.4 V for Fe_2_O_3_, and −0.6 V for MoO_3_. Furthermore, Figure 4 reveals not only the decrease of the overpotential for CO_2_ reduction, but also the higher faradaic currents for the Fe_2_O_3_ electrode, being the best electrocatalytic electrode under the experimental conditions. It should be noted that differences in electrode interface area may vary and are desirable for proper electrode performance comparison.

On the other hand, TMOs, such as Fe_2_O_3_ and MoO_3_, are n-type semiconductors, and their catalytic activity could be related to the conduction band position [32]. The conduction band position could estimate if the electron transfer is thermodynamically feasible. Although several conduction and valence band position diagrams are found in literature for the thin film electrodes studied, these values correspond to conditions different from those applied in this study, and are generally under vacuum conditions [33]. However, the potential at which the conduction band is found depends on factors such as the type of oxide, charge state of the surface, presence of adsorbed species, pH, and isoelectric point of the material [34]. In this way, the flat band potential was estimated via photoelectrochemical measurements in the presence of low oxidation potential species such as methanol [35]. This potential should be very close to the potential of the conduction band lower edge, and related with the apparition of photocurrents. Appendix A shows the LSV profile for Fe_2_O_3_ and MoO_3_ electrodes under transitory illumination, in 0.1 M de TBAF/ACN/+3.0 M CH_3_OH, purged with N_2_. The results show that photocurrent onset potentials start at −0.70 V for Fe_2_O_3,_ and 0.65 V for MoO_3_. Therefore, the electron transference from the TMO to CO_2_ molecule and the formation of thermodynamically favored C_1_ products from CO_2_ reduction are feasible. Figure 1 represents the conduction band position for TMO under the conditions of this study, and the potentials (versus SHE at pH 7) for some relevant CO_2_ redox couples (Figure 1).

The results show a generalized catalytic behavior for CO_2_ electrocatalytic reduction over the TMO electrodes used. In all cases, a decrease in the overpotential related to that required for the first CO_2_ electron transfer is evident. According to the dissociative reduction model, in the presence of CO_2_, the formation of OV caused by the application of increasingly negative potentials appears to be favored by the stabilization of Fe^2+^ and Mo^5+^ species upon CO_2_ chemisorption. Therefore, it is plausible to think that CO_2_ could be adsorbed in reduced metal sites, and that the OV generation is favored. Metal reduction is facilitated, causing an increase in capacitive currents in CV profiles, even at potentials where CO_2_ reduction does not proceed. In this sense, it is reasonable to think that the catalytic surface of the examined TMO materials possesses a different morphology and adsorption energy for CO_2_, and, therefore, a different activation capability [11,36,37]. Likewise, Tanabe et al. found that basic metal oxides facilitate the transfer of electrons to CO_2_ molecules adsorbed on the surface by adduct formation [38].

Although the activation of the CO_2_ molecule by adsorption on the surface of the catalyst favors its reduction, it is not the only factor that determines the electrocatalytic activity of the electrode. The formation of the adsorbed surface does not always lead to CO_2_ reduction [36]. In this sense, it is important to consider the need of electron-rich active sites giving rise to electron transfer from the TMO surface to the CO_2_ molecule. Thus, the formation of reduced sites identified as catalytic sites for the reduction of CO_2_ occurs [21,39]. Density Functional Theory calculations show that OV with n-type semiconductor behavior enhance the surface state near the conduction band [40]. Based on the results obtained, the increment of surface states improves the electrode catalytic activity even at potentials at which CO_2_ reduction does not occur.

Finally, conduction band position, OV density and their associated energy, as well as surface state density are parameters needed to understand the electrocatalytic behavior of TMO materials. Table 1 presents the initial flat band and faradaic current potentials, Ef_b_ and Ef_i_, for the electrocatalytic reduction of CO_2_.

The initial faradaic onset potential for Fe_2_O_3_ electrodes is less negative than those at the the conduction band position. This fact reveals a high density of surface states near the conduction band; from which, electron transfer to the adsorbed CO_2_ molecules occurs. In the case of MoO_3_, Ef_b_ and Ef_i_ are very close to each other. The initial potential for MoO_3_ is 0.05 V, which is less negative than its flat band potential. This could be related to the fact that the metal Mo has different oxidation numbers; therefore, the excess charge from the formation of OV can be distributed on both surface and bulk atoms, causing the excess charge to delocalize, and the density of electron-rich sites to decrease. The migration capability for OV in TMO such as TiO_2_ and WO_3_ is predicted by DFT studies, as well as (0 0 1)-orientation activation energies of 0.76 V and 0.14 V, respectively [41,42]. This means that excess electrons resulting from OV formation in TMO such as MoO_3_ may migrate to the bulk structure more seamlessly than Fe_2_O_3_ [43], which suggests a greater availability of Lewis basic active sites on the TMO surface for CO_2_ reduction when the OV migration energy is higher.

## 3. Materials and Methods

The preparation of Fe_2_O_3_ and MoO_3_ electrodes was carried out by chemical bath deposition and the sol-gel method, respectively [28]. Experimental details for oxide thin-film synthesis and the characterization are presented in the Appendix A.

A computer-controlled micro-Autolab potentiostat was used for all electrochemical experiments. A conventional three-electrode glass cell was used with an Ag/AgCl(s)/KCl (aq,sat) reference electrode and a Pt wire as a counter electrode in all the experiments. The supporting electrolyte was 0.1 M tetrabutylammonium perchlorate (TBAP) (≥99.0% from Sigma-Aldrich, Madrid, Spain) in ACN (99.88% anhydrous from Scharlau, Multisolvent, HPLC grade, Barcelona, Spain). In all the experiments, the electrolyte was purged for 30 min with either N_2_ (Alphagaz 99.999%, Madrid, Spain) or CO_2_ (Air Liquid, 99.98%, Madrid, Spain). A conversion of potentials measured to SHE scale is the same as previously reported, and according to Pavlishchuk and Addison. Detailed information about materials, methods, and the potential conversion scale are presented in the Appendix A.

## 4. Conclusions

Based on the results obtained, TMOs present an electrocatalytic behavior for the reduction of CO_2_ in ACN medium. In this sense, it could be affirmed that the location of the conduction band, the ability to generate OV, and charge accumulation in surface states are factors that can modify the charge transfer from TMO to CO_2_. The results obtained show that, in general terms, the less negative (in potential) the conduction band position is, the more positive the potential at which the processes of faradic reduction of CO_2_ are initiated. This behavior is expected, since there is a correlation between the location of the edge of the conduction band and the electronic surface states responsible for the CO_2_ activation. In this sense, future studies that allow obtaining information on the generation energy and the migration of OVs may be relevant for the evaluation of a TMO as a possible (photo)cathode for CO_2_ reduction. Finally, it should be noted that carbon monoxide was obtained as a reduction product, which is in accordance with what is expected for aprotic systems.

## Data Availability

Not applicable.

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
