# Peer review of "Insights of Fe2O3 and MoO3 Electrodes for Electrocatalytic CO2 Reduction in Aprotic Media"

_ijms, 2022, doi:10.3390/ijms232113367_

Round 1
Reviewer 1 Report
The research topic is interesting but the manuscript itself is not adding much. Already Abstract is too general and does not present any significant achievement while it should be seen as the shopping window. The first two sentences belong to Introduction but not to Abstract.
In short, the English is dreadful and at places hard to understand. For example on p.5: ..decrease of overpotential related with the needed for first CO2 electron transfer.. - this sentence really does not give sense. On p3., Additionally,..an additional peak...shows up. The terminology, syntax and orthography are hardly acceptable.
Also, careful editing is required. In the affiliations, which author belong to index 3? The phD degree = PhD. Acetonitrile abbreviations (ACN) sometimes appear as CAN (Figure 3). Page 5, line 155 refer to Eq.3 but the real Eq.3 shows something different. Subscripts and superscripts in References are not formatted. Flat bland in Table 1 should read flat band.
Most importantly, the presented science contains some puzzles:
- Which product is obtained by the electrocatalytic reduction of CO2? How many electrons are consumed? Equations 3, 4 and 4b do not reveal anything. What are the TON values and faradaic efficiency? Is there any mechanistic insight (oxygen abstraction at 'oxygen vacancies')? How can CO2 stabilize reduced TMO sites is there is a catalytic electron transfer reaction?
- What experimental method is square-wave chronoamperometry in Figure 4?
- The title announces aprotic conditions but methanol (CH3OH) was added in the photoelectrochemical measurements. Explain clearly the reason. What is 'low oxidation potential species'?
- The supporting electrolyte in Experimental is tetrabutylammonium perchlorate, TBAP. Thus, no TBAP+, as presented on p.3. However, the TBAF abbreviation occurs at many places in the text, without the explanation of F. Further, TBA+ are bulky, not voluminous.
Minor points:
- Several reference electrode potentials appear in the texts = SHE, Ag/Ag+, and Ag/AgCl used by the authors. It would be beneficial (in Experimental) to compare the values, using a single reference.
- In the manuscript, 'more positive' potential shifts should be changed to 'less negative', in line with the negative signs.
Concluding, there is much work to be done to convert the current manuscript to a publishable version.
Author Response
We are grateful to this reviewer for his/her relevant comments. Regarding the reviewer’s specific comment, we want to present the following considerations:
see the attached word file

Reviewer 2 Report
The paper is well presenred, I recommend it for publication with only one minor remark
on figure 1 it should be interpreted and depicted directly into figure with what process are all signals connected. especially anodic contrapeaks
Author Response
We are grateful to this reviewer for his/her positive comments. Regarding the reviewer’s specific comment, we want to present the following considerations:
see the attached word file

Reviewer 3 Report
The authors investigated CO2 reduction of Fe2O3 and MoO3 in aprotic solvents. Overall, its boring to read the paper. Suggested the following modifications.
Need references for Acetonitrile being used as non-aqueous solvent in CO2 reduction.
Authors needs to connect in the introduction section why WO3 and TiO2 are specifically used why not MnO2 or other transition metal oxides. The specialties of WO3 and Fe2O3. This connects the work with the existing research.
Line 84…. Singly?
Line 91 OV formation... what is OV?
There are a lot of reactions happening based on results and discussion first few paragraphs. Its better to indicate the most possible reactions on each peak in the Figure. 1.
What is the faradaic efficiency of each electrode?
Line 171 – what is OMTs? When using abbreviation please define when you use for the first time.
Authors described on various places about the surface states, LUMO, Orbitals etc… But with no literature support. A lot of description is hard to understand. Its strongly suggested giving a pictorial representation based on the existing literature.
All figures are of very poor quality.
How is thin film catalysts' activity compared to powder nanoparticles? Can you compare it with the literature?
Author Response

(The authors gave the same response as above.)

Round 2
Reviewer 1 Report
Compared to the original version, the revised manuscript has improved significantly. The experiments are described correctly, e.g., the double-potential step chronoamperometry and the analytical detection of CO in the gas phase. Given the expanding field of TMO-catalysed reduction of CO2 I recommend publishing this contribution.
Prior to publishing, however, I still recommend careful reading and correcting minor but frequent typos and inaccurate formulations. For example:
1/ p.3, .127 - Mo+6, Mo+5, Fe+3 should be Mo6+, Mo5+, Fe3+.
2/ Caption to Figure 2: Black and red correspond to N2 and CO2, individually.
Better: The black and red curves correspond to scans under an atmosphere of N2 and CO2, respectively.
3/ Table 1 - Name the reference electrode for the potentials given in the table. Note that Figure and Tables must be independent of the main text, which regards also the captions, headings, footnotes, etc.
Correct 'flat bland' in the Table caption to 'flat band'.
4/ The correct spelling of the name Palishchuk in the main text and supporting Supporting Information is Pavlishchuk (see also ref. 2 in Supporting Information).
Finally, I propose to mention the established electrocatalytic generation of CO also in Conclusions.
Author Response
Response to the comments of the reviewer
Insights of Fe2O3 and MoO3 electrodes for electrocatalytic CO2 Reduction in Aprotic Media
Néstor E. Mendieta-Reyesa, Alejandra Sophia Lozano Péreza, Carlos Alberto Guerrero Fajardo*a
a Departamento de Química, Facultad de Ciencias, Universidad Nacional de Colombia, Cra 30 # 45-03, Edificio 451, Bogotá, Colombia.
*Corresponding Author: caguerrerofa@unal.edu.co
Taking into account the comments of the reviewer 1:
“Compared to the original version, the revised manuscript has improved significantly. The experiments are described correctly, e.g., the double-potential step chronoamperometry and the analytical detection of CO in the gas phase. Given the expanding field of TMO-catalysed reduction of CO2 I recommend publishing this contribution.”
We are grateful to this reviewer for his/her positive comments and the recommendation for this paper to be published.
“Prior to publishing, however, I still recommend careful reading and correcting minor but frequent typos and inaccurate formulations. For example:
1/ p.3, .127 - Mo+6, Mo+5, Fe+3 should be Mo6+, Mo5+, Fe3+.”
Following the reviewer’s comment, in figure 1 line 127 p.3, said typos where corrected and rewritten.
“2/ Caption to Figure 2: Black and red correspond to N2 and CO2, individually.
Better: The black and red curves correspond to scans under an atmosphere of N2 and CO2, respectively.”
Regarding the reviewer’s comment, the caption to figure 2 was corrected as suggested.
“3/ Table 1 - Name the reference electrode for the potentials given in the table. Note that Figure and Tables must be independent of the main text, which regards also the captions, headings, footnotes, etc.
Correct 'flat bland' in the Table caption to 'flat band'.”
The comment was accepted and the reference electrode information (Ag/AgCl/KCl(sat)) was added in the caption to table 1. In line 270, the word flat band was corrected.
“4/ The correct spelling of the name Palishchuk in the main text and supporting Supporting Information is Pavlishchuk (see also ref. 2 in Supporting Information).”
The misspelling of the name of the researcher Pavlishchuk was corrected in both, the main text (Line 301) and in the supporting information.
“Finally, I propose to mention the established electrocatalytic generation of CO also in Conclusions”
The recommendation of the reviewer was included in the conclusion as a remarkable product of the paper.
Reviewer 3 Report
The authors have revised the manuscript according to the reviewer's comments and the responses are satisfactory. Therefore the paper can be accepted in the present form.
Author Response
Response to the comments of the reviewer
Insights of Fe2O3 and MoO3 electrodes for electrocatalytic CO2 Reduction in Aprotic Media
Néstor E. Mendieta-Reyesa, Alejandra Sophia Lozano Péreza, Carlos Alberto Guerrero Fajardo*a
a Departamento de Química, Facultad de Ciencias, Universidad Nacional de Colombia, Cra 30 # 45-03, Edificio 451, Bogotá, Colombia.
*Corresponding Author: caguerrerofa@unal.edu.co
Taking into account the comments of the reviewer:
"The authors have revised the manuscript according to the reviewer's comments and the responses are satisfactory. Therefore the paper can be accepted in the present form."
We are grateful to this reviewer for his/her positive comments and the recommendation for this paper to be published. Additionally, moderate English changes were done.